# The Effect of Drying–Wetting Cycles on Soil Inorganic Nitrogen Transformation in Drip-Irrigated Cotton Field Soil in Northwestern China

Honghong Ma [1,2,3], Zhiying Yang [1,2], Shenghai Pu [1,2,3] and Xingwang Ma [1,2,*]

1 Institute of Soil, Fertilizer and Agricultural Water Conservation, Xinjiang Academy of Agricultural Sciences, Urumqi 830091, China
2 Key Laboratory of Northwest Oasis Agriculture Environment, Ministry of Agriculture, Urumqi 830091, China
3 Xinjiang Huier Agriculture Group Co., Ltd., Changji 830011, China
* Correspondence: maxw@xaas.ac.cn

**Abstract:** Drip irrigation under plastic mulch is widely used and leads to periodic drying–rewetting (DW) cycles in Xinjiang, Northwest China. However, the effect of different wet and dry alternation types on soil inorganic nitrogen transformation is not clear. Studying these issues not only provides reference for the formulation of fertilization and irrigation systems but is also of great significance for reducing non-point source pollution. An incubation experiment was conducted in 2018 in Baotou Lake farm in Korla City, Xinjiang, with drip-irrigated cotton (*Gossypium hirsutum* L.). The treatments were designed comprising three parts: (1) DW intensity (Q100, Q90, Q80, Q70, Q60); (2) DW frequency (P3d, P5d, P7d, P9d, P11d); and (3) soil wetting time (P1m, P3m, P5m). The results revealed that the contents of the $NH_4^+$ and $NO_3^-$ decreased with the increase in the DW intensity and were highest in the Q100 treatment. The rate of net N mineralization decreased with the increase in the DW intensity. The highest rate (7.02 mg $kg^{-1}$ $d^{-1}$) was found in the Q70 treatment in the wet to dry process and 3.03 mg $kg^{-1}$ $d^{-1}$ in the Q60 treatment in the dry to wet process, respectively. The contents of the $NH_4^+$ and $NO_3^-$ were higher with the higher DW frequency (P3d). The rate of net N mineralization decreased with the increase in the DW frequency and was highest in the P3d treatment in the wet to dry process and the P5d treatment in the dry to wet process, respectively. The soil wetting time was longer with the content of $NH_4^+$, $NO_3^-$, and N mineralization (P5m). The rate of net N mineralization was higher with the longer soil wetting time in the wet to dry process and the shorter soil wetting time from the dry to wet process. These results demonstrate that a reasonable DW intensity, DW frequency, and soil wetting time could not only effectively promote nitrogen transformation and the absorption of nitrogen but also reduce the nitrogen losses under drip irrigation.

**Keywords:** frequency; intensity; mineralization; transformation; wetting time

## 1. Introduction

Global extreme climate change, with both drought and precipitation, occurs frequently, which causes intensive changes in the hydrological cycles at both global and regional scales [1]. Such phenomena occur regularly in arid and semiarid areas [2,3]. Nitrogen (N) is often a limiting macronutrient in many territorial systems, and its transformation is largely driven by fluctuations in the soil moisture, such as drying–rewetting (DW) [4]. Soil DW can significantly affect the soil microbial activity and alter the N transformation, which has attracted great attention during the last few decades in the context of extreme climate change [5]. Soil organic matter (SOM) is an important pool of organic N. It is mineralized to simple inorganic forms by microbial communities [6]. Soil organic N mineralization is the most significant process, which determines the absorption of N available to plants and is influenced by abiotic factors [7]. Soil water content is one of the most important abiotic factors, which is easily influenced by the soil DW and affects the N availability, from organic

N to inorganic N [7]. Interestingly, laboratory studies of N transformation have quantified wetting–drying cycles in recent decades [8]. These studies mainly focused on wetland soil, grassland soil, and forest soil, while there has been less research on agricultural soil [9–11]. Agricultural soil plays an important role in national production and life and should receive great attention. Cotton (*Gossypium hirsutum* L.) is one of the most important cash crops in Xinjiang autonomous province, located in northwestern China, an arid region [12]. Most of the cotton plant types in Xinjiang use drip fertigation. Affected by the small amount of single irrigation and high frequency of drip irrigation, the soil experiences periodic alternation between dry and wet during cotton growth.

The rate of nitrogen mineralization in the soil increased after the DW cycles compared with that kept moist [5]. DW cycles promoted N mineralization in alpine wetlands, while nitrogen mineralization decreased after repeated wet and dry conditions [13,14]. During the soil DW cycle, plants and microbes can have different sensitivities to soil moisture, which may cause temporal shifts and delays in the net N mineralization by microbes and the N uptake by plants, with long-term implications for N loss and plant productivity [15]. This excitation effect may be facilitative, inhibitory, or ignored during the DW cycle, and the mechanisms are still unclear.

The objectives of this study were: (1) to define the dynamics of nitrate nitrogen ($NO_3^-$—N) and ammonium nitrogen ($NH_4^+$—N) in different DW cycle treatments under drip irrigation in an arid area; (2) to explore the characteristics of N mineralization from the wet to dry and dry to rewet processes under drip irrigation in an arid area; and (3) to demonstrate which treatment benefits N transformation under drip irrigation in an arid area. The results of the study will be particularly helpful for reasonable water and nitrogen management.

## 2. Materials and Methods

### 2.1. Soil Sampling

The soils were sampled from a cotton field, located at the long-term positioning monitoring base on Baotou Lake farm in Korla City, Xinjiang, China (41.6903 N, 85.8719 E). The soils were collected in the beginning of spring (between April and May) in 2018. The crops were cotton (*Gossypium hirsutum* L.). The area has a continental arid climate with an annual precipitation of 56.2 mm and evaporation of 2497.4 mm. The accumulated temperature above 10 °C and the frost-free period are 4252.2 °C and 205 days, respectively. The groundwater level is 2 m to 2.5 m. The soil in the field is classified as a sandy loam soil in the Chinese soil classification, and it has medium fertility. The bulk density of the surface soil (0–30 cm) in the field is 1.23 g cm$^{-3}$.

The soils were collected from 0 to 30 cm deep in the H horizon with a soil auger. After the soil was naturally dried, the debris, such as stones, cotton straw, plastic mulch, and roots, were removed by hand. The soil was immediately brought to the laboratory and passed through a 2 mm sieve. The soil contained 7.51 g kg$^{-1}$ organic matter, 0.45 g kg$^{-1}$ total N, 0.046 g kg$^{-1}$ total P, 95.93 mg kg$^{-1}$ available potassium (K), 7.27 mg kg$^{-1}$ $NO_3^-$—N, and 3.61 mg kg$^{-1}$ $NH_4^+$—N, had a pH of 8.46, and had a bulk density of 1.57 g cm$^{-3}$.

### 2.2. Incubation Experiment

The sieved soil was homogenized, and 100 g soil dry weight was placed in a 100 mL disposable plastic cup (Figure 1). The soil water content was adjusted to 60% of the soil water-holding capacity (WHC), then preincubated for seven days at room temperature (about 25 °C) in the dark. We chose 60% WHC as the dry treatment, because it is the critical value for cotton growth in Xinjiang.

The incubation experiment included three experiments: dry and wet alternating strength, dry and wet alternating frequency, and soil wetting time.

For the dry and wet alternating strength experiment, five treatments were set: 100–60% WHC (Q100), 90–60% WHC (Q90), 80–60% WHC (Q80), 70–60% WHC (Q70), and 60% WHC (Q60). The soil water content was 60% WHC as the control treatment (CK). For each

treatment, the soil water content was adjusted to 100% WHC, 90% WHC, 80% WHC, 70% WHC, and 60% WHC, as the initial water content, respectively. When the water content of each treatment became 60%, deionized water was added to reach 100% WHC, 90% WHC, 80% WHC, and 70% WHC once more, respectively. Each treatment was incubated for two dry–wet cycles.

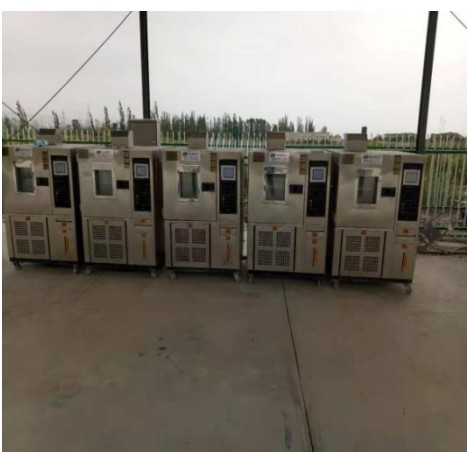 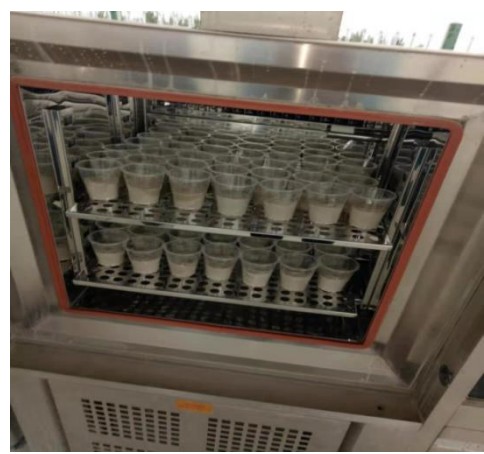

**Figure 1.** Incubation experiment.

For the dry and wet alternating frequency experiment, five treatments were set: 3d (P3d), 5d (P5d), 7d (P7d), 9d (P9d), and 11d (P11d). All the water contents were adjusted to 100% WHC as the initial water content for each treatment. For the 3d treatment, the deionized water was added on the third day. For the 5 d treatment, the deionized water was added on the fifth day. The other treatments followed this method. Each treatment was incubated for two dry–wet cycles.

For the soil wetting time experiment, three treatments were set: 1 m (P1m), 3 m (P3m), and 5 m (P5m). All the water contents were adjusted to 100% WHC as the initial water content for each treatment. For the 1 m treatment, the deionized water was added in 1 min by controlling the adding rate. The other treatments followed this method. Each treatment was incubated for two dry–wet cycles. When the soil water content was 60% WHC in each treatment, the deionized water was added to reach 100% WHC. The second incubation cycle began. Each treatment was incubated for two dry–wet cycles.

All disposable cups were placed in a 25 °C constant temperature incubation (Hermostatic climate chamber RP-250 °C, China).

### 2.3. Sampling Method

Before the experiment, 72 replicates of the same treatment were prepared in order to simulate two drying–rewetting cycles. Then, three replicates of each treatment were randomly selected daily during the incubation period. Then, the soil samples were collected to determine the soil $NO_3^-$—N and $NH_4^+$—N. The sampling time was two alternating dry and wet processes.

### 2.4. Chemical Analyses

The soil $NO_3^-$—N and $NH_4^+$—N were determined with a Continuous Flowing Analyzer (Bran Luebbe, Germany) in 2.0 mol $L^{-1}$ KCl extracts [16]. The net N nitrification (Nm) was calculated as follows:

$$N_m = [(NO_3^-—N + NH_4^+—N)_a - (NO_3^-—N + NH_4^+—N)_b]/T.$$

Note: $N_m$ represents the net N mineralization (mg $kg^{-1}$ $d^{-1}$); a represents the ending of the incubation experiment; b represents the beginning of the incubation experiment; and T represents the number of incubation days.

### 2.5. Statistical Analysis

Differences in the nitrate nitrogen ($NO_3^-$—N), ammonium nitrogen($NH_4^+$—N), and N mineralization among days in different treatments were analyzed by one-way analysis of variance using SPSS version 11.5 (SPSS, Chicago, IL, USA). Line charts were prepared to assess the changes in the$NO_3^-$—N and $NH_4^+$—N in different treatments. Bar charts were prepared to assess changes in the N mineralization of different process in different treatments.

## 3. Results

### 3.1. $NO_3^-$ and $NH_4^+$ Dynamics during the DW Strength Experiment

The content of the $NO_3^-$obtained its maximum value at the moment of adding the deionized water in the first DW cycle. In the second DW cycle, the content of the $NO_3^-$decreased with the increase in the incubation time. The content of the $NO_3^-$ in the Q100 treatment was 72.92 mg kg$^{-1}$ and 69.17 mg kg$^{-1}$ in the first and second DW cycles, respectively. The content of the $NO_3^-$ in the Q90 treatment was 67.26 mg kg$^{-1}$ and 64.50 mg kg$^{-1}$ in the first and second DW cycles, respectively. The content of the $NO_3^-$ in the Q80 treatment was 61.83 mg kg$^{-1}$ and 58.74 mg kg$^{-1}$ in the first and second DW cycles, respectively. The content of the $NO_3^-$ in the Q70 treatment was 69.04 mg kg$^{-1}$ and 60.36 mg kg$^{-1}$ in the first and second DW cycles, respectively. The content of the $NO_3^-$ in the Q60 treatment was 65.44 mg kg$^{-1}$ and 62.88 mg kg$^{-1}$ in the first and second DW cycles, respectively. (Figure 2A). Overall, the content of the $NO_3^-$ was the maximum in the Q100 treatment during the whole DW cycle incubation. The content of the $NO_3^-$ in the first DW cycle was higher than that in the second DW cycle. During the two-cycle incubation, the Q100 treatment had a 7.27%, 15.15%, 8.93%, and 9.69% higher average $NO_3^-$ compared to the Q90, Q80, Q70, and Q60 treatments, respectively.

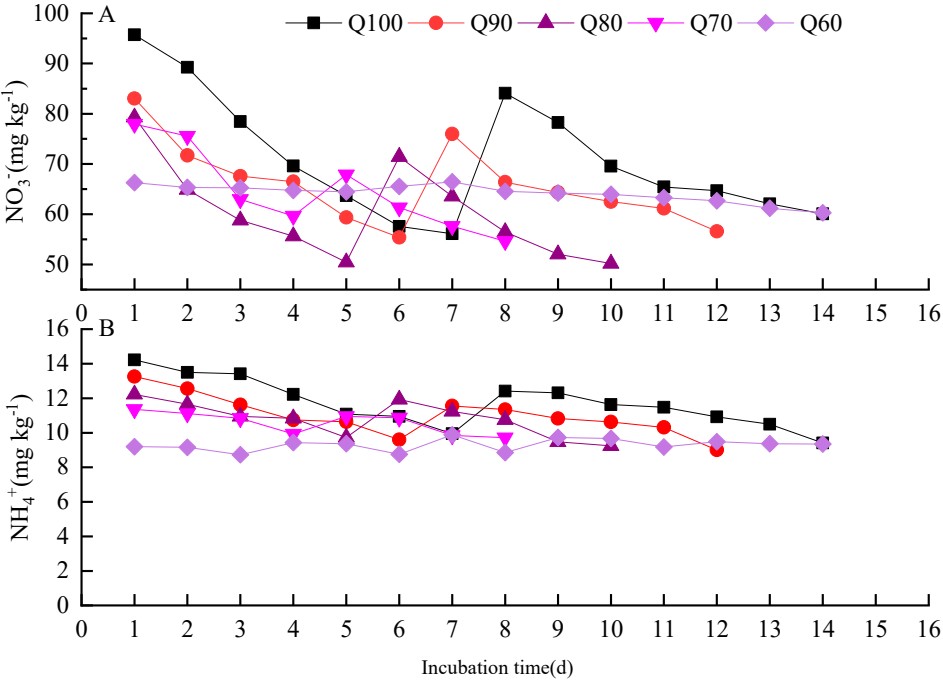

**Figure 2.** The contents of the $NO_3^-$ and $NH_4^+$ in different dry and wet alternating strengths. Note: (**A**,**B**) in the left corner represent the contents of the $NO_3^-$ and $NH_4^+$, respectively.

The content of the $NH_4^+$ obtained its maximum value at the moment of adding the deionized water in the first DW cycle. In the second DW cycle, the content of the $NH_4$ decreased with the increase in the incubation time (Figure 2B). The content of the $NH_4^+$ in the Q100 treatment was 12.19 mg kg$^{-1}$ and 11.23 mg kg$^{-1}$ in the first and second DW

cycles, respectively. The content of the $NH_4^+$ in the Q90 treatment was 11.40 mg kg$^{-1}$ and 10.61 mg kg$^{-1}$ in the first and second DW cycles, respectively. The content of the $NH_4^+$ in the Q80 treatment was 11.08 mg kg$^{-1}$ and 10.53 mg kg$^{-1}$ in the first and second DW cycles, respectively. The content of the $NH_4^+$ in the Q70 treatment was 10.81 mg kg$^{-1}$ and 10.35 mg kg$^{-1}$ in the first and second DW cycles, respectively. The content of the $NH_4^+$ in the Q60 treatment was 9.22 mg kg$^{-1}$ and 9.37 mg kg$^{-1}$ in the first and second DW cycles, respectively. Overall, the content of the $NH_4^+$ was at the maximum in the Q100 treatment. The content of the $NH_4^+$ in the first DW cycle was higher than that in the second DW cycle. During the two-cycle incubation, the Q100 treatment had a 6.03%, 7.79%, 9.70%, and 20.66% higher average $NH_4^+$ compared to the Q90, Q80, Q70, and Q60 treatments, respectively.

### 3.2. $NO_3^-$ and $NH_4^+$ Dynamics during the DW Frequency Experiments

The content of the $NO_3^-$ obtained its maximum value at the moment of adding the deionized water in the first and second DW cycle. Then, the content of the $NO_3^-$ decreased with the increase in the incubation time (Figure 3A). The content of the $NO_3^-$ in the P3d treatment was 66.40 mg kg$^{-1}$ and 69.85 mg kg$^{-1}$ in the first and second DW cycles, respectively. The content of the $NO_3^-$ in the P5d was 61.38 mg kg$^{-1}$ and 61.16 mg kg$^{-1}$ in the first and second DW cycles, respectively. The content of the $NO_3^-$ in the P7d treatment was 59.50 mg kg$^{-1}$ and 58.55 mg kg$^{-1}$ in the first and second DW cycles, respectively. The content of the $NO_3^-$ in the P9d was 56.69 mg kg$^{-1}$ and 56.62 mg kg$^{-1}$ in the first and second DW cycles, respectively. The content of the $NO_3^-$ in the P11d treatment was 54.64 mg kg$^{-1}$ and 54.42 mg kg$^{-1}$ in the first and second DW cycles, respectively. Overall, the content of the $NO_3^-$ was at the maximum in the P3d treatment. The content of the $NO_3^-$ in the first DW cycle was higher than that in the second DW cycle. During the two-cycle incubation, the P3d treatment had a 10.07%, 13.36%, 16.84%, and 19.95% higher average $NO_3^-$ compared to the P5d, P7d, P9d, and P11d treatments, respectively.

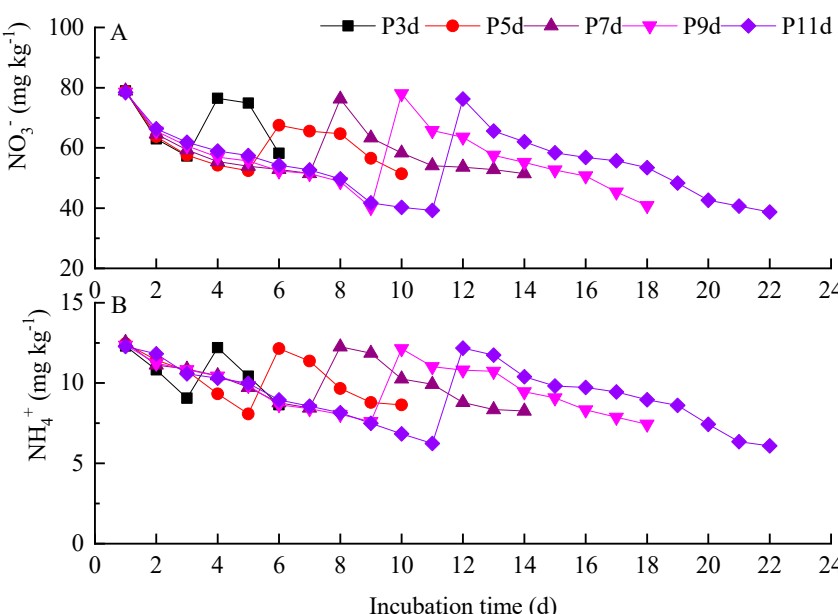

**Figure 3.** The contents of the $NO_3^-$ and $NH_4^+$ in different dry and wet alternating frequencies. Note: (**A**,**B**) in the left corner represent the contents of the $NO_3^-$ and $NH_4^+$, respectively.

The trends of the $NO_3^-$ and $NH_4^+$ were consistent. The content of the $NH_4^+$ obtained the maximum value at the moment of adding the deionized water in the first and second DW cycle. Then, the content of the $NH_4^+$ decreased with the increase in the incubation time (Figure 3B). The content of the $NH_4^+$ in the P3d treatment was 10.73 mg kg$^{-1}$ and 10.42 mg kg$^{-1}$ in the first and second DW cycles, respectively. The content of $NH_4^+$ in the P5d treatment was 10.41 mg kg$^{-1}$ and 10.12 mg kg$^{-1}$ in the first and second DW

cycles, respectively. The content of $NH_4^+$ in the P7d treatment was 10.28 mg kg$^{-1}$ and 9.95 mg kg$^{-1}$ in the first and second DW cycles, respectively. The content of the $NH_4^+$ in the P9d treatment was 9.71 mg kg$^{-1}$ and 9.65 mg kg$^{-1}$ in the first and second DW cycles, respectively. The content of $NH_4^+$ in the P11d treatment was 9.20 mg kg$^{-1}$ and 9.16 mg kg$^{-1}$ in the first and second DW cycles, respectively. Overall, the content of the $NH_4^+$ was at its maximum in the P3d treatment. The content of the $NH_4^+$ in the first DW cycle was higher than that in the second DW cycle. During the two-cycle incubation, the P3d treatment had a 2.92%, 4.36%, 8.46%, and 13.22% higher average $NH_4^+$ compared to the P5d, P7d, P9d, and P11d treatments, respectively.

### 3.3. $NO_3^-$ and $NH_4^+$ Dynamics during the Soil Wetting Time Experiment

The content of the $NO_3^-$ in the P1m treatment was 44.38 mg kg$^{-1}$ and 49.25 mg kg$^{-1}$ in the first and second DW cycles, respectively (Figure 4A). The content of the $NO_3^-$ in the P3m treatment was 59.63 mg kg$^{-1}$ and 57.82 mg kg$^{-1}$ in the first and second DW cycles, respectively. The content of the $NO_3^-$ in the P5m treatment was 65.28 mg kg$^{-1}$ and 59.79 mg kg$^{-1}$ in the first and second DW cycles, respectively. Overall, the content of the $NO_3^-$ was at its maximum in the P5m treatment. The content of the $NO_3^-$ in the first DW cycle was higher than that in the second DW cycle. During the two-cycle incubation, the P5m treatment had a 19.19% and 6.09% higher average $NO_3^-$ compared to the P1m and P3m treatments, respectively.

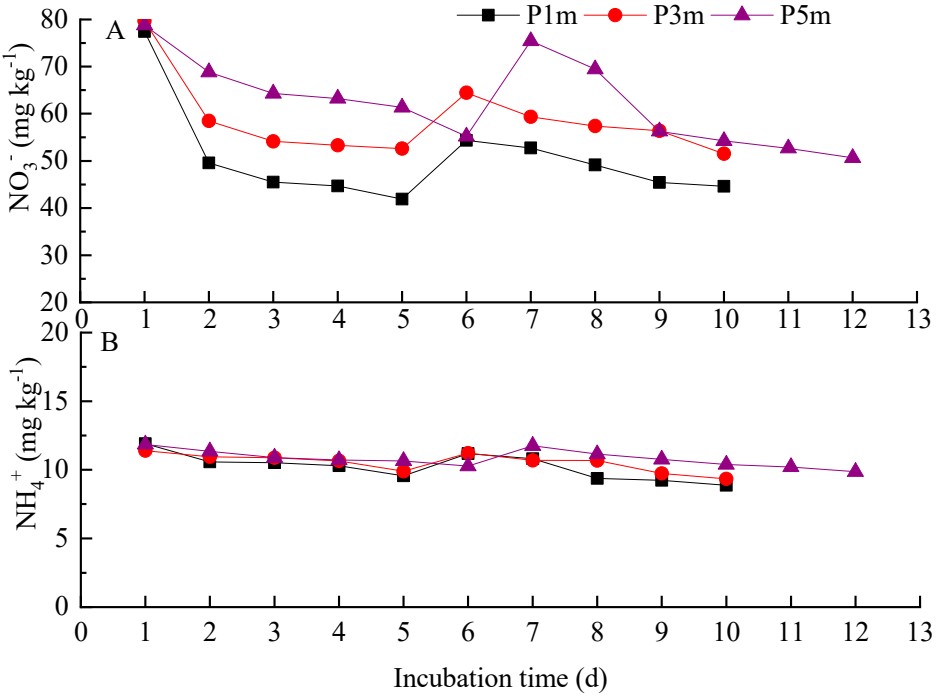

**Figure 4.** The contents of the $NO_3^-$ and $NH_4^+$ in different dry and wet alternating frequencies. Note: (**A**,**B**) in the left corner represent the contents of the $NO_3^-$ and $NH_4^+$, respectively.

The content of the $NH_4^+$ in the P1m treatment was 10.39 mg kg$^{-1}$ and 9.89 mg kg$^{-1}$ in the first and second DW cycles, respectively (Figure 4B). The content of the $NH_4^+$ in the P3m treatment was 10.75 mg kg$^{-1}$ and 10.33 mg kg$^{-1}$ in the first and second DW cycles, respectively. The content of the $NH_4^+$ in the P5m treatment was 10.95 mg kg$^{-1}$ and 10.68 mg kg$^{-1}$ in the first and second DW cycles, respectively. Overall, the content of the $NH_4^+$ was at its maximum in the P5m treatment. The content of $NH_4^+$ in the first DW cycle was higher than that in the second DW cycle. During the whole incubation period, the change in the $NH_4^+$ was relatively more stable than the $NO_3^-$. During the two-cycle

incubation, the P5m treatment had a 5.44% and 2.58% higher average $NH_4^+$ compared to the P1m and P3m treatments, respectively.

### *3.4. Net N Mineralization*

In the wet to dry process, the net N mineralization was positive in the Q60 treatment, which indicated the N mineralization was greater than the N fixation (Figure 5A). In the dry to wet process, the net N mineralization was negative in these treatments, which indicated the N fixation was greater than the N mineralization (Figure 5B). In addition, the net N mineralization in the P7d and P5m treatments in the wet to dry process was at the maximum, which indicated the N fixation was higher than in the rest of the treatments. The net N mineralization in the Q60, P5d, and P1m treatment in the dry to wet process was at the maximum, which indicated the N fixation was higher than in the rest of the treatments.

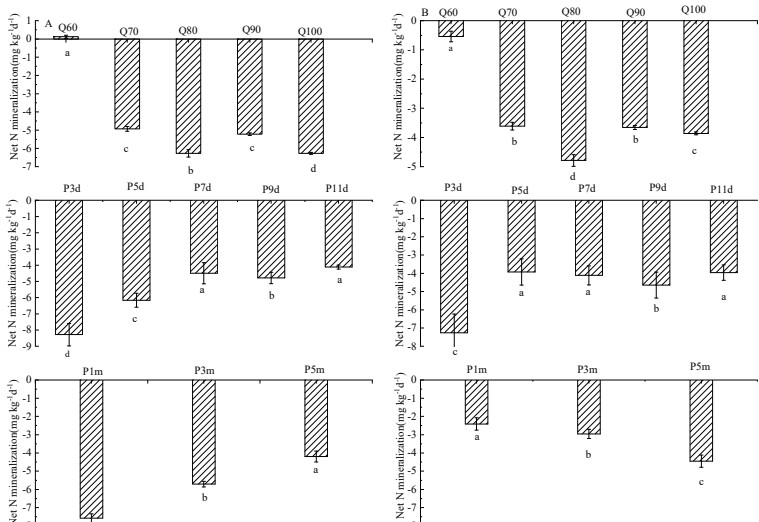

**Figure 5.** The net soil N mineralization in different treatments. Note: (**A**) represents the net soil N mineralization in the wet to dry process; (**B**) represents the net soil N mineralization in the dry to wet process. Different lowercase letters indicate significant differences among the treatments at the 0.05 level.

### 4. Discussion

### *4.1. N Transformation*

Noah Fierer and Joshua P. Schimel (2002) found that the concentrations of $NH_4^+$ in oak and grass soil were very low and were not affected by the drying–rewetting cycle treatments [17]. In our experiment, the contents of the $NO_3^-$ and $NH_4^+$ were lower in the second drying–rewetting cycle than in the first drying–rewetting cycle in all treatments, while the contents of the $NO_3^-$ and $NH_4^+$ sharply increased when the water was supplied. These results are consistent with Hu [18], mainly because the drying–rewetting cycles can enhance microbial activity, and more nitrogen is consumed [19]. Miller et al. (2005) found that N fixation and net N mineralization were higher with the lower DW frequency [20]. In our DW frequency experiment, the contents of the $NO_3^-$ and $NH_4^+$ in the P3d treatment were significantly higher than in other treatments. This result was consistent with Miller et al. [20]. For the soil wetting time treatment, the contents of the $NO_3^-$ and $NH_4^+$ in the P5m treatment were significantly higher than in the P1m and P3m treatments, probably because the soil moisture in the P1m and P3m treatments changed rapidly, resulting in flooded soil with low oxygen, which inhibited the nitrogen mineralization. Soil aerobic microorganisms were severely water-stressed as time passed, resulting in the death of a large number of microorganisms, and the surviving microorganisms or new anaerobic microorganisms could not fully utilize the newly formed or exposed N sources in time after rewetting, leading to a decrease in the content of the $NH_4^+$. Then, the content of the $NH_4^+$ increased, probably because the microorganisms were more resilient after sev-

eral DW cycles, and the increase in the microbial population reversed the decline in the $NH_4^+$ [21–23].

*4.2. N Mineralization*

The N mineralization was initially higher due to the short-term fluctuations in the DW cycle. After a period of time, the rates of nitrogen mineralization decreased and reached equilibrium [24]. The N mineralization was higher in the Q60 treatment (Figure 1), probably due to the high N requirement for microorganisms at this suitable soil moisture. This confirmed our hypothesis that litter can increase microbial N immobilization, but this process can be modulated by soil moisture.

Microorganisms play a key role in the mineralization of soil organic N, which could be broken down into inorganic N and absorbed by plants [25,26]. In our experiment, the rate of net N mineralization was negative in most DW treatments, which indicated the DW cycle was beneficial for N fixation. Agehara and Warncke (2005) found that nitrification, denitrification, and ammonification were affected by the soil moisture, microbial activity, microbial community structure, and the rate of diffusion of the soluble substrates in organic matter [27]. When the content of the soil moisture was in the range of 45–75% of the water holding capacity, the number and activity of ammonia-oxidizing bacteria increased with the increase in the content of the soil moisture, which positively contributed to mineralization [28,29]. In our experiment, the content of the $NO_3^-$ was higher when the soil water content was 60% of the water holding capacity, while the $NH_4^+$ was higher when the soil water content was in the range of 60–100% of the water holding capacity. The nitrogen mineralization significantly decreased with the increase in the nitrogen application in agricultural soil. In our experiment, the rate of N mineralization did not show regular rules in different DW intensities, but the rate of N mineralization showed a positive value in the Q80, P5d, P9d, and P5m treatments, suggesting that the N mineralization was greater than the N fixation and beneficial for cotton growth and nutrient absorption. In our study, although the characteristics of the inorganic nitrogen in the DW intensity, DW frequency, and soil wetting time of the incubation experiments were explored, the mechanism that causes the transformation is not yet clear; so, it remains an urgent problem to be solved and desirable to study.

## 5. Conclusions

Cotton, one of the main cash crops, has widely adopted drip irrigation under plastic film in Xinjiang, Northwest China, which leads the soil to experiencing periodic drying–rewetting (DW) cycles. The incubation experimental results showed that the contents of the $NO_3^-$ and $NH_4^+$ were lower in the second cycle than in the first cycle in all treatments, except the Q60 treatment. For the different DW treatments, the content of the $NO_3^-$ and $NH_4^+$ was higher with the greater DW strength, the higher DW frequency, and the longer soil wetting time. In the wet to dry process, the N mineralization was greater than the N fixation in the Q60 treatment. In the dry to wet process, the N fixation was greater than the N mineralization in all treatments. These results imply that too much or too little irrigation, a lower irrigation frequency, and a higher drip head flow are detrimental to the soil inorganic nitrogen transformation, which may also be an important reason for the failure to increase the cotton yield.

**Author Contributions:** H.M. contributed to the conceptualization, methodology, investigation, data collection, acquisition framework, data analysis, and writing the original draft manuscript in terms of the scientific content. Z.Y. and S.P. contributed to the conceptualization, methodology, investigation, and design of the experiment. X.M. contributed to the conceptualization, data collection, acquisition framework, supervision, and design of the experiment. All authors have read and agreed to the published version of the manuscript.

**Funding:** This research was funded by Basic Research Business Fund for Public Welfare Research Institutes in the Autonomous Region (KY2022028), Youth Science in the Autonomous Region (2022D01B165), and Project of Fund for Stable Support to Agricultural Sci-Tech Renovation (xjnkywdzc-2022005).

**Institutional Review Board Statement:** Not applicable.

**Informed Consent Statement:** Not applicable.

**Data Availability Statement:** Research Data Policies at https://www.mdpi.com/ethics (accessed on 12 February 2023).

**Acknowledgments:** The contents are the sole responsibility of the authors. Thanks to Dongmei Zhao, Yongfeng Tu and Jisheng Yue for their support of the experiment.

**Conflicts of Interest:** The authors declare no conflict of interest.

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
