# Peer review of "The Effect of Drying–Wetting Cycles on Soil Inorganic Nitrogen Transformation in Drip-Irrigated Cotton Field Soil in Northwestern China"

_applsci, doi:10.3390/app13063892_

Round 1

Reviewer 1 Report

This manuscript needs major revision before being accepted for publication. The detailed comments are listed as follows:

A.    Title of the manuscript

The title of the manuscript looks like a sentence rather than a title. Please rename it.

B.      Abstract

Line 13 and 14, “However, it is not clear how nitrogen transformed and the characteristic of inorganic nitrogen is under different DW treatments?” Please rewrite the sentence with a full stop rather than question mark.

C.    Section 2.2. Incubation experiment

Please provide images of the soil before and after incubation.

D.    Section 3.4. Net N mineralization

“N fixation” and “N fixcation”, are they the same thing or just a typo?

E.     Statistical analyses

The manuscript only presents the charts and describe the results using comparative adjectives. How large is the exact difference? Please statistically analyse the results by calculating the percentage difference between those treatments.

F.     Schematic diagram

Schematic diagram(s) is/are highly encouraged to show the workflow of the experiments.

G.    English expression

The English grammar and expression should be professionally checked. Typos should be eliminated.

Author Response

  1. Title of the manuscript.The title of the manuscript looks like a sentence rather than a title. Please rename it.

Reply: The editor’s concern is greatly appreciated. Followed your valuable suggestion, I reconsider the title of this manuscript, and revise the title: Effect of Drying-rewetting Cycles on Soil Inorganic Nitrogen Transformation in Drip Irrigated Cotton Field Soil in North-western China

  1. Abstract

Line 13 and 14, “However, it is not clear how nitrogen transformed and the characteristic of inorganic nitrogen is under different DW treatments?” Please rewrite the sentence with a full stop rather than question mark.

Reply: The editor’s concern is greatly appreciated. Followed your valuable suggestion, I reconsider the sentence of this manuscript, and revise the sentences as follows: the effect of different wet and dry alternation types on soil inorganic nitrogen transformation is not clear.

  1. Section 2.2. Incubation experiment

Please provide images of the soil before and after incubation

Reply: The editor’s suggestion is greatly appreciated! I have provided the images of incubation experiment(Figure 1)

Figure 1.Incubation experiment

  1. Section 3.4. Net N mineralization

“N fixation” and “N fixcation”, are they the same thing or just a typo?

Reply: The editor’s suggestion is greatly appreciated! I am sorry for this error. I rechecked the word “N fixation”.

  1. Statistical analyses

The manuscript only presents the charts and describe the results using comparative adjectives. How large is the exact difference? Please statistically analyse the results by calculating the percentage difference between those treatments.

Reply: The editor’s suggestion is greatly appreciated! Followed your valuable suggestion, I analyze the results by calculating the percentage difference between these treatments as follows:

3.1 During the 2 cycle incubation, the Q100 treatment had 7.27%, 15.15%, 8.93%, 9.69% higher average NO3- compared to the Q90, Q80, Q70 and Q60 treatment, respectively.

During the 2 cycle incubation, the Q100 treatment had 6.03%, 7.79%, 9.70%, 20.66% higher average NH4+ compared to the Q90, Q80, Q70 and Q60 treatment, respectively.  

3.2 During the 2 cycle incubation, the P3d treatment had 10.07%, 13.36%, 16.84%, 19.95% higher average NO3- compared to the P5d, P7d, P9d and P11d treatment, respectively.  

During the 2 cycle incubation, the P3d treatment had 2.92%, 4.36%, 8.46%, 13.22% higher average NH4+ compared to the P5d, P7d, P9d and P11d treatment, respectively.

3.3 During the 2 cycle incubation, the P5m treatment had 19.19%, 6.09% higher average NO3- compared to the P1m and P3m treatment, respectively.  

During the 2 cycle incubation, the P5m treatment had 5.44%, 2.58% higher average NH4+ compared to the P1m and P3m treatment, respectively.  

  1. Schematic diagram

Schematic diagram(s) is/are highly encouraged to show the workflow of the experiments.

Reply: The editor’s suggestion is greatly appreciated! I will use more schematics diagram(s) to show the work-flow of the experiments in future.

  1. English expression

The English grammar and expression should be professionally checked. Typos should be eliminated.

Reply: The editor’s suggestion is greatly appreciated! I am sorry for this error. I have checked the English grammar and expression.

Reviewer 2 Report

The authors have done an excellent job by studying nitrogen transformation in drying-rewetting treatments. The research is quite interesting. However, although it deals with an important issue, there is not much information or data about nitrifying bacteria. Unfortunately, this may degrade the value of the manuscript. The second issue is the study was conducted in only one location. It could have been repeated over time or conducted in two locations to consider the different environmental conditions.

Some minor corrections are required before publication.

Author Response

1.Please replace it with "revealed"

Reply: The editor’s concern is greatly appreciated. I have replaced “found” with “revealed”.

2. areas

Reply: The editor’s concern is greatly appreciated. I have revised it.

3. Please provide reference(s)

Reply: The editor’s suggestion is greatly appreciated! I have provided reference.

4. followed

Reply: The editor’s concern is greatly appreciated. I have revised it.

5.Please avoid using the same phrases

Reply: The editor’s suggestion is greatly appreciated! I am sorry for this error. I have rechecked the relevant information, and deleted “mean”.

6.which microorganism actually do you refer to? Please, clarify

Reply: The editor’s suggestion is greatly appreciated! Microorganism actually refer to microbes. I have changed it in the manuscript.

7.treatments.

Reply: The editor’s suggestion is greatly appreciated! I am sorry for this error. I have added the punctuation.

8.I agree that the soil microbial community plays a crucial role in nitrification.

It would increase the quality of the manuscript if the authors isolated and identified nitrifying bacteria as well as determining the change in the amount of these bacteria for each treatment.

Reply: The editor’s concern is greatly appreciated. Followed your valuable suggestion, I will mention the soil microbial community in the next manuscript in order to explain the mechanism of inorganic nitrogen.

Reviewer 3 Report

In the Material & Methods section:

2.2. Incubation experiment: 

Have any extra doses of N been added as a standard to the treatments? or just the amount initially contained in the collected soil (0.45 g kg-1 total N)?

Could this temperature of incubation (25 °C) have influenced the N transformations? In this region, the soil temperature in the cotton growing season is probably much higher than 25 °C.

2.3. Sampling method:

Please, would it be possible to better explain how these 3 replicates were taken? Were they taken from the same amount of initial soil or were they 3 different soil collections?

In the latter case would it be replications? 

Please detail that.

The CONCLUSIONS section needs to be rewritten. In the Conclusions section, we can find many phrases that are already in the Results section. I suggest that the authors be direct in the conclusions, clearly pointing out the contribution of the manuscript to science and how these scientific findings can influence the cultivation of cotton or other plants in the region under study. They could also point out indications for future studies to be carried out based on the results achieved.

Author Response

  1. That word is already in the title. Please replace it.

Reply: The editor’s concern is greatly appreciated. Followed your valuable suggestion, I have replaced it with mineralization.

  1. put a space here

Reply: The editor’s concern is greatly appreciated. I am sorry for this error.I have revised it.

  1. I suggest that the authors' names be removed from this paragraph. Suggestion: DW cycles can promote N mineralization in alpine wetlands while nitrogen mineralization decreased after repeated wet and dry conditions[14,15].

Reply: The editor’s suggestion is greatly appreciated! I have revised this sentence.    

  1. Have any extra doses of N been added as a standard to the treatments? or just the amount initially contained in the collected soil (0.45 g kg-1 total N)?

Reply: The editor’s suggestion is greatly appreciated! Just the amount initially contained in the collected soil (0.45 g kg-1 total N) has been as a standard to the treatments.

  1. and

Reply: The editor’s suggestion is greatly appreciated! I am sorry for this error. I have added “and” in the sentences.

  1. for

Reply: The editor’s suggestion is greatly appreciated! I am sorry for this error. I have added “for” in the sentences.

  1. for

Reply: The editor’s suggestion is greatly appreciated! I am sorry for this error. I have added “for” in the sentences.

  1. For

Reply: The editor’s suggestion is greatly appreciated! I am sorry for this error. I have added “for” in the sentences.

  1. For

Reply: The editor’s suggestion is greatly appreciated! I am sorry for this error. I have added “for” in the sentences.

  1. Could this temperature have influenced the N transformations? In this region, the soil temperature in the cotton growing season is probably much higher than 25 °C.

Reply: The editor’s suggestion is greatly appreciated! One of the students in our group measured the soil temperature from 25 to 27 ℃in 0-20 cm during the cotton growth stage. Thus 25 °C was chosen as incubation temperature.

  1. Please, would it be possible to better explain how these 3 replicates were taken? Were they taken from the same amount of initial soil or were they 3 different soil collections?In the latter case would it be replications? Please detail that.

Reply: The editor’s suggestion is greatly appreciated! Maybe this question confused other readers. 3 replicates were taken from the same amount of initial soil.Before the experiment, 72 replicates of the same treatment were prepared in order to simulate 2 drying-rewetting cycles, especially for P11d treatment. Thus, I took 24 duplicate samples from each treatment in order to facilitate statistics. Then according to the different treatment design, I added different amount of deionized water to simulate different DW strength, DW frequency and soil wetting time.  

  1. include a space here.

Reply: The editor’s suggestion is greatly appreciated! I am sorry for this error. I have rechecked it.

  1. Include a point here.

Reply: The editor’s suggestion is greatly appreciated! I am sorry for this error. I have revised it.

  1. .

Reply: The editor’s suggestion is greatly appreciated! I am sorry for this error. I have revised it.

  1. r

Reply: The editor’s suggestion is greatly appreciated! I am sorry for this error. I have revised it.

  1. Suggestion to rewrite this paragraph: These results are consistent with Hu[19], mainly because drying-rewetting cycles can make substrate available for microorganisms, and more N was consumed or fixed due to enhanced microbial activity and increased biomass[20].

Reply: The editor’s suggestion is greatly appreciated! I am sorry for this error. I have revised it as follows: These results are consistent with Hu[19], which is mainly because drying-rewetting cycles could enhance microbial activity and more nitrogen is consumed[20].

  1. include a space here

Reply: The editor’s suggestion is greatly appreciated! I am sorry for this error. I have revised it.

  1. showed a

Reply: The editor’s suggestion is greatly appreciated! I am sorry for this error. I have revised it.

  1. Be

Reply: The editor’s suggestion is greatly appreciated! I am sorry for this error. I have revised it.

  1. d

Reply: The editor’s suggestion is greatly appreciated! I am sorry for this error. I have revised it.

  1. This CONCLUSIONS section needs to be rewritten.In this Conclusions section we can find many phrases that are already in the Results section.I suggest that the authors be direct in the conclusions, clearly pointing out the contribution of the manuscript to science and how these scientific findings can influence the cultivation of cotton or other plants in the region under study. They could also point out indications for future studies to be carried out based on the results achieved.

Reply: The editor’s suggestion is greatly appreciated! I have revised them as follows:

Cotton, one of the main cash crops, is widely adopted drip irrigation under plastic film in Xinjiang, Northwest China, which leaded the soil to experiencing periodic drying-rewetting(DW) cycle. The incubation experiments results showed that the content of NO3- and NH4+ were less in the second cycle than that in the first cycle in all treatments, except Q60 treatment. For different DW treatments, the content of NO3- and NH4+ was higher with the greater DW strength, higher DW frequency and longer soil wetting time. In the wet to dry process, N mineralization was greater than N fixation under Q60 treatment. In the dry to wet process, N fixation was greater than N mineralization in all treatments.These results imply that too much or too little irrigation, the lower irrigation frequency and the higher drip head flow, are detrimental to soil inorganic nitrogen transformation, which may also be an important reason for the failure to increase cotton yield.

Round 2

Reviewer 1 Report

The revised manuscript is acceptable for publication.